# Milk Fat Globule Membrane Relieves Fatigue via Regulation of Oxidative Stress and Gut Microbiota in BALB/c Mice

**DOI:** 10.3390/antiox12030712

**Published:** 2023-03-13

**Authors:** Xiaoxiao Zou, Wallace Yokoyama, Xiaohui Liu, Kai Wang, Hui Hong, Yongkang Luo, Yuqing Tan

**Affiliations:** 1Key Laboratory of Functional Dairy, College of Food Science and Nutritional Engineering, China Agricultural University, Beijing 100083, China; 2Healthy Processed Foods Research Unit, Agricultural Research Service, United States Department of Agriculture, Albany, CA 94710, USA; 3Department of Product and Development, Hebei Dongkang Dairy Co., Ltd., Shijiazhuang 052165, China

**Keywords:** milk fat globule membrane, anti-fatigue, antioxidant, gut microbiome

## Abstract

Milk fat globule membranes (MFGMs) are complex structures that incorporate bioactive proteins and lipids to assist in infant development. However, the antifatigue and antioxidant potentials of MFGM have not been investigated. In this study, repeated force swimming measured fatigue in male BALB/c mice fed MFGM and saline for 18 weeks. The MFGM supplementation increased the time to exhaustion by 42.7% at 6 weeks and 30.6% at 14 weeks (*p* < 0.05). Fatigue and injury-related biomarkers, including blood glucose, lactic acid, and lactate dehydrogenase, were ameliorated after free swimming (*p* < 0.05). The activity of antioxidant enzymes in blood serum increased at 18 weeks, while malondialdehyde (MDA) content decreased by 45.0% after the MFGM supplementation (*p* < 0.05). The Pearson correlation analysis showed a high correlation between fatigue-related indices and antioxidant levels. The increased protein expression of hepatic Nrf2 reduced the protein expression of Caspase-3 in the gastrocnemius muscle (*p* < 0.05). Moreover, the MFGM supplementation increased the relative abundance of *Bacteroides*, *Butyricimonas*, and *Anaerostipes*. Our results demonstrate that MFGM may maintain redox homeostasis to relieve fatigue, suggesting the potential application of MFGM as an antifatigue and antioxidant dietary supplement.

## 1. Introduction

Fatigue is a symptom that accompanies various diseases, such as hypertension, depression, diabetes, and coronary heart disease. Long-term accumulated fatigue leads to chronic fatigue syndrome (CFS), including symptoms such as muscle pain and weakness [1]. Proposed mechanisms of physical fatigue include exhaustion of energy stores, clogging by exercise metabolites, and excessive free radicals [2]. The exhaustion theory emphasizes the dramatic consumption of energy storage molecules, such as ATP, glucose, and fat, during exercise, resulting in the declining performance of skeletal muscle and failure to complete a predetermined exercise intensity, thus generating physical fatigue [3]. The clogging theory suggests that excessive accumulation of harmful metabolites, for instance, blood lactic acid (BLA), during strenuous exercise results in energy supply obstruction and muscle capacity reduction [4]. In recent years, the free radical fatigue theory has attracted more interest. This theory postulates that intense exercise increases free radicals, which oxidize fatty acids in cell membranes and damage protein and nucleic acid chains. The free radical damage puts the body in a state of oxidative stress, thereby giving rise to physical fatigue [5]. The intestinal tract is a crucial source of reactive oxygen species (ROS) and nitrogen oxide species [6]. Meanwhile, it has been declared that after the supplementation of antioxidant foods, altered intestinal bacteria promote the release of anthocyanins or polyphenols from food to increase the antifatigue capacity of the body [7]. Therefore, fatigue is greatly associated with the homeostatic control of the redox intestinal environment. It has been observed that active ingredients derived from natural plants and animals can reduce fatigue, with the merit of low side effects [8].

Milk fat globule membrane (MFGM) has been demonstrated to possess various nutritional functions and is recognized as the sole source of phospholipids in breast milk [9]. It has been demonstrated that formula supplemented with MFGM can reduce the neurodevelopmental differences between breast milk and formula feeding in postnatal rat pups [10]. A previous study found that MFGM promoted the growth of beneficial intestinal bacteria, inhibited the colonization of pathogenic bacteria, and increased the content of short-chain fatty acids [11]. In addition, MFGM improved body weight, fasting blood glucose, and serum insulin levels in type 2 diabetic (T2D) mice [12]. In healthy adult mice, MFGM plus exercise increased Peroxisome proliferator-activated receptor gamma coactivator 1-alpha (*Pgc1α*) expression by increasing adiponectin production in, or secretion from, adipose tissues, thereby activating fatty acid oxidation and, subsequently, improving endurance capacity [13]. These results indicated that MFGM has significant nutritional functions and may be a promising bioactive ingredient for foods. While various functional properties of MFGM have been studied, their antioxidant and antifatigue functions have not yet been well studied.

Hence, this study aims to investigate the effects of MFGM on the antifatigue capacity of BALB/c mice and the effect of MFGM feeding on the gut microbiome. We hypothesized that MFGM exerts its antifatigue function by relieving oxidative stress.

## 2. Materials and Methods

### 2.1. Materials

MFGM (Hilmar 7500) was donated by Hilmar Cheese Company (Hilmar, CA, USA). The protein, fat, ash, and moisture contents of MFGM were 66.47%, 14.98%, 2.75%, and 5.79%, respectively. Protein, fat, ash, and moisture were measured according to the AOAC International methods 955.04, 2003.06, 942.05, and 934.01, respectively. The carbohydrate of MFGM was 10.01%, which was calculated by deducting the contents of protein, moisture, fat, and ash from 100% according to the AOAC 979.06 (AOAC International, 2012) [14].

#### 2.1.1. Amino Acids of MFGM

The amino acid composition was measured by the method of Li et al. with minor modifications using HPLC-MS/MS analysis [15]. The HPLC-MS/MS experiments were performed using an Ultimate 3000 (Dionex, Sunnyvale, CA, USA)-API 3200 Q TRAP detector (AB Sciex, Framingham, MA, USA), an HPLC system, and an MSLab50AA-C18 column (150 mm × 4.6 mm × 5 um). The sample was pretreated as follows: The MFGM sample was homogenized with distilled water. Then, the homogenate was added to an equal volume of concentrated (37%) hydrochloric acid and digested at 110 °C for 21 h under nitrogen. The digested solution was membrane filtered, dried under a vacuum, and reconstituted with distilled water. Finally, the sample was derivatized with ITRAQ reagent (Applied Biosystems, Waltham, MA, USA) after sonication for 5 min. The remainder of the analysis followed Li’s method. The composition of amino acids in the MFGM used in this study is shown in Appendix A.

#### 2.1.2. Phospholipids of MFGM

The MFGM sample was extracted with acetone to remove triglycerides. Phospholipids were extracted with chloroform and methanol (3:2). The phospholipids were analyzed using HPLC by elution through an LC-NH2 column (25 cm × 4.6 mm, Sigma, St. Louis, MO, USA). The phospholipids were quantified by an evaporative light scattering detector (Infinity 1260 II, Agilent, Santa Clara, CA, USA) [16]. The total and specific composition of phospholipids in the MFGM sample are displayed in Table 1.

### 2.2. Animal Experiment

The experiments were approved by the Biomedical Ethics Committee of Peking University, Beijing, China (Protocol code: LA2021477). Male BALB/c mice were purchased from Charles River Development, Inc. (Beijing, China) at 7 weeks old. The mice were housed in a plastic case under a temperature of 22 ± 2 °C and a humidity of 50–70%, and operating on a 12 h light–12 h dark cycle with food and water available ad libitum. The maintenance feed for the mice was obtained from Beijing Keao Xieli Feed Co., Ltd. (Beijing, China), which contained more than 18% protein and 4% fat. The contents of leucine, valine, and isoleucine (compositions of branched-chain amino acids) in the feed were 14.80 g, 8.90 g, and 7.40 g per 1 kg, respectively (http://www.keaoxieli.com/product/137.html, accessed on 10 March 2023). The mice were randomly assigned into 2 groups (8 mice per group). One group was orally administered MFGM (MFGM group) at 400 mg/kg body weight (BW). The control group was orally administered an equal amount of saline (Control group). After 6 and 14 weeks of feeding, an exhaustive swimming exercise test was performed. At 16 weeks, a free-swimming test was performed, and blood was collected. The mice were sacrificed after 18 weeks of feeding. Blood, liver, gastrocnemius muscle, and other tissues were collected for further analysis. The detailed procedures are shown in Appendix A.

### 2.3. Exhaustive Swimming Exercise Test

The exhaustive swimming exercise test was modified from the study by Chen et al. [17]. A weight of 7% that was equivalent to a mouse’s body weight was hung on each mouse’s tail. The mice were administered either MFGM or saline 30 min before the swimming test. The mice were placed in a tank with water at a depth greater than 30 cm at 25 ± 1.0 °C. The mice swam until exhaustion, which was defined as a time point when the mice failed to return to the water surface to breathe for 7 s. The time period from the beginning of the test to the endpoint was recorded as the exhaustive swimming time.

### 2.4. Free Swimming Test

At 16 weeks, the mice were submitted to a free-swimming test without any load. The mice were administered saline or MFGM 30 min before the swimming test. The mice were placed in a circular tank (36 × 36 cm), and after 60 min of swimming, blood samples were collected via the submandibular vein. The blood serum was separated by centrifugation before storage at −80 °C.

### 2.5. Organ and Tissue Collection

The mice were sacrificed via an intraperitoneal injection of 50 mg/kg pentobarbital sodium after 18 weeks of feeding. The mice were fasted for 8 h to minimize the possible impacts of uncertain time of food intake. The liver, heart, thymus, and spleen were weighed and stored at −80 °C.

About 1 mL of blood was collected from the retro-orbital sinus, refrigerated for 1 h at 4 °C, and centrifuged at 1180× *g* for 15 min at 4 °C. The blood serum was stored at −80 °C for subsequent analysis. The gastrocnemius muscle from the left leg was fixed in a 4% paraformaldehyde solution for tissue microscopy and immunocytochemical analysis. The gastrocnemius muscle from the right leg and liver was stored at −80 °C for other analysis.

### 2.6. Serum Biochemical Indices

The levels of blood glucose (BG), blood lactic acid (BLA), and lactate dehydrogenase (LDH) were measured after the free-swimming test. BLA, superoxide dismutase (SOD), catalase (CAT), glutathione peroxidase (GSH-Px), and malondialdehyde (MDA) were determined after sacrificing the mice. These assay methods were performed according to the methods from the Nanjing Jiancheng Bioengineering Institute (Nanjing, China).

### 2.7. Histological Analysis

Gastrocnemius muscles were fixed overnight in a 4% paraformaldehyde solution and then embedded in paraffin. The paraffin sections stained with hematoxylin and eosin (H&E) were viewed under a bright field on an upright optical microscope (Nikon Eclipse E100, Tokyo, Japan). The slices from the two groups were selected with the same area of view.

### 2.8. Immunohistochemical Analysis

The immunochemical method was used to determine the protein expression level of Caspase-3 in gastrocnemius muscles [18]. Cleaved Caspase-3 rabbit polyclonal antibody and HRP-conjugated goat anti-rabbit IgG (H + L) were provided by Wuhan Servicebio Technology Co., Ltd. (Wuhan, China). ImageJ 1.53k (National Institution of Health, Bethesda, MD, USA) was used to measure the expression level.

### 2.9. Western Blot Analysis

Total proteins were isolated from the liver tissue using a cell lysis buffer for Western and IP (Beyotime Biotechnology Co., Ltd., Shanghai, China). The nuclear and cytoplasmic proteins were extracted through using a Nuclear and Cytoplasmic Protein Extraction Kit (Beyotime Biotechnology Co., Ltd., Shanghai, China). Protein content was quantified by a BCA Protein Assay Kit (Beijing Solarbio Science & Technology Co., Ltd., Beijing, China). The amount of protein in each group was equalized to 20 ug and then subjected to sodium dodecyl sulfate–polyacrylamide gel electrophoresis. The fractionated proteins were transferred to polyvinylidene fluoride (PVDF) membranes in WIX-miniBLOT4 (WIX Technology Co., Ltd., Beijing, China). The protein-rich PVDF membranes were blocked with 5% skim milk-TBS solution for 1.5 h, and later incubated overnight at 4 °C with the following primary antibodies: Nuclear factor erythroid-derived 2-like 2 (Nrf2), Kelch-like ECH-associated protein 1 (Keap1; 1:1200), and β-actin (1:1000) were all purchased from Beijing Biosynthesis Biotechnology Co., Ltd., Beijing, China. Horseradish peroxidase (HRP)-conjugated second antibodies (Beyotime; 1:1000 in 5% skim milk-TBS solution) were added and incubated at room temperature for 1.5 h. The density of protein bands was analyzed by using the Quantity One 1-D Analysis Software (Bio-Rad Laboratories, Inc., Hercules, CA, USA) and ImageJ 1.53k (National Institution of Health, Bethesda, MD, USA).

### 2.10. 16S rRNA Gene Sequencing and Bioinformatic Analysis

The feces of the mice, which were mixed in pairs, were collected at 18^th^ week and were immediately stored at −80 °C until further use. The method of 16S rRNA gene sequencing was performed according to Li et al. [19], with minor modifications. DNA was extracted from the feces according to the instruction of the E.Z.N.A.^®^ soil DNA Kit (Omega Bio-Tek, Norcross, GA, USA). The V3–V4 domains of 16S rRNA were amplified using primers 338F (5’-ACTCCTACGGGAGGCAGCAG-3’) and 806R(5’-GGACTACHVGGGTWTCTAAT-3’) by an ABI GeneAmp^®^ 9700 PCR thermocycler (Applied Biosystems, Foster City, CA, USA). Based on the standard protocols developed by Majorbio Bio-Pharm Technology Co., Ltd. (Shanghai, China), purified amplicons were pooled into equimolar and paired-end sequences on an Illumina MiSeq PE300 platform/NovaSeq PE250 platform (Illumina, San Diego, CA, USA). The data analysis and mapping were performed on the online platform of Majorbio Cloud Platform (www.majorbio.com, accessed on 10 March 2023) and Bioinformatics (http://www.bioinformatics.com.cn/, accessed on 10 March 2023). The Kyoto Encyclopedia of Genes and Genomes (KEGG) analysis, according to the result of the 16S rRNA gene sequencing, was performed on the online platform of Majorbio Cloud Platform. Firstly, PICRUSt was used to standardize the OTU abundance table. Secondly, the corresponding KEGG Ortholog (KO) information of OTU was obtained through the green gene id corresponding to each OTU, and each KO’s abundance was calculated. Finally, the plots were configured on https://www.omicsolution.org/wkomics/main/ (accessed on 10 March 2023). The raw data were deposited into the NCBI Sequence Read Archive (SRA) database with the accession number PRJNA891493.

### 2.11. Statistical Analysis

All data are shown as means ± SD (standard deviation). Statistical analysis was determined by independent-sample *t*-test. Statistical significance was established at *p* < 0.05 level. These calculations were carried out via the SPSS software (version 25.0, IBM Inc., Chicago, IL, USA). The correlation between fatigue-related and antioxidant indices was revealed using a Pearson correlation analysis, and the plot was constructed on https://omicsolution.org/wkomics/main/ (accessed on 10 March 2023).

## 3. Results and Discussion

### 3.1. Animal Metrics and Structure of Muscle Tissue

The liver, heart, thymus, and spleen indices (the ratio of the organ to body weight) after 18 weeks of feeding are shown in Figure 1A,B. The body weights and organ indices are not different after MFGM supplementation. Haramizu et al. also reported no differences in body weight, liver weight, or heart weight in BALB/c mice fed MFGM for 12 weeks [13]. These findings confirm that MFGM supplementation is nutritious and safe.

As revealed in Figure 1C,D, compared to the Control group, the gastrocnemius muscle cells in the MFGM group are arranged more tightly and orderly, representing irregular polygons, while the shapes are circular in the Control group. The intercellular space is also smaller after MFGM supplementation.

Li and coworkers reported that milk fat globule-EGF factor 8 (MFG-8), alternatively known as lactadherin, and accounting for 82.35% of MFGM protein, promoted C2C12 cell proliferation via the phosphatidylinositol 3-kinase and mammalian target of rapamycin (PI3K/Akt/mTOR/P70S6K) signaling pathway [20]. A previous study found that dietary MFGM combined with exercise could improve endurance performance through increased lipid metabolism. It is noteworthy that sphingomyelin (SM) may be one of the critical factors in increasing *Pgc1α* mRNA expression in soleus muscle in vivo and differentiating myoblasts in vitro [13]. Increasing cells’ regenerative potential and proliferation activity can relieve sarcopenia. These reports suggest that the ingredients of MFGM, such as MFG-8 and SM, may regulate gene expressions and metabolism in muscle cells to improve gastrocnemius muscle structure, ultimately strengthening the capacity for antifatigue and endurance.

### 3.2. Swimming Test and Serum Biomarkers of Antifatigue Status

As shown in Figure 2A,B, there are significant increases in the exhaustive swimming time (EST) at not only 6 weeks (42.69% higher, *p* < 0.05) but also 14 weeks (30.60% higher, *p* < 0.05) after MFGM supplementation.

Several biochemical indices derived from the blood serum were analyzed to quantify the energy reserve and the accumulation of detrimental metabolites after intense exercise. In this study, the BG levels of the mice fed MFGM increased at both 16 weeks and 18 weeks (Figure 2C,D, *p* < 0.05). BG is the primary energy source for the central nervous system. A lack of BG during exercise can cause hypoglycemia, followed by fatigue [21].

As displayed in Figure 2E,F, both BLA and LDH are lower in the mice in the MFGM group by 25.68% and 18.93%, respectively, compared to the Control group. During high-intensity exercise, the move from aerobic metabolism to anaerobic glycolysis or glycogenolysis leads to a massive accumulation of BLA [1]. The reduced pH caused by LA can affect cardiac circulation and skeletal muscle system function [22]. LDH is a major enzyme involved in anaerobic glycolysis and gluconeogenesis by promoting the redox reaction between lactic acid and pyruvic acid that reduces the accumulation of BLA. However, the cell membrane becomes more permeable under vigorous exercise, resulting in the escape of LDH into the blood, thereby reducing the enzymatic oxidation of LA [23]. The lower levels of LDH and BLA in the MFGM group compared to the levels in the Control group after the free-swimming test are consistent with the results of previous studies.

The tryptophan (Trp)-5-hydroxytryptamine (5-HT)-central fatigue hypothesis proposes that the synthesis and release of the neurotransmitter 5-HT by neurons may lead to central fatigue after exercise. Trp and 5-HT levels increase during exercise. The rate-limiting and critical step of this procedure is the transport of Trp across the blood–brain barrier [24]. Because branched-chain amino acids (BCAAs, including leucine, valine, and isoleucine) are transported via the same carrier system, decreasing the free Trp/BCAA ratio can delay central fatigue [25]. The MFGM used in this study contained a significant amount of BCAAs, including 6.7% of Val, 8.3% of Leu, and 4.7% of Ile (Appendix A). Falavigna and coworkers found that diets that were chronically supplemented with BCAAs had a beneficial effect on performance by sparing glycogen in the soleus muscle and inducing a lower concentration of plasma ammonia in rats, therefore leading to increased performance in the swimming test [26]. BCAAs may be the main MFGM component responsible for the antifatigue effects observed in this study.

### 3.3. Serum Biomarkers of Antioxidant Enzymes

Superoxide dismutase (SOD), catalase (CAT), and glutathione peroxidase (GSH-Px) concentrations of the mice fed MFGM are 6.71%, 87.1%, and 14.42%, respectively, which are higher than those of the mice in the Control group (Table 2, *p* < 0.05). The MFGM diet also lowers the content of malondialdehyde (MDA) in the mice fed this diet compared to the Control group (Table 2, *p* < 0.05).

SOD, CAT, and GSH-Px are the primary endogenous antioxidant enzymes that can protect the body from molecular free radical damage caused by excessive oxidative stress by scavenging reactive oxygen species (ROS) and their metabolites [27]. Under normal conditions, ROS is effectively eliminated by antioxidant defense systems. Growing evidence demonstrates that vigorous exercise can break the oxidation/antioxidant balance. Overproduction of free radicals can damage biological macromolecules, including proteins, lipids, and DNA, in the contracting myocytes and influence cellular signaling pathways [27]. Finally, muscle fatigue happens. Free radicals act on lipids to cause peroxidation reactions, and MDA is the final product of oxidation. The concentration of MDA is a crucial biomarker of oxidative injury in the body.

MFGM antioxidant peptides were partially purified, and their antioxidant activity was measured. The peptide fraction with a molecular weight of 5–10 ku had the greatest antioxidant activity. Their DPPH clearance was 70.41% or 0.8 times that of vitamin C [28]. Li et al. identified two novel antioxidant peptides, TGIIT and YAR, from MFGM hydrolysates. These antioxidant peptides had pro-proliferative activity and protected rat myoblasts against dexamethasone-induced oxidative damage and apoptosis by regulating mitochondrial activity and inhibiting mitochondria-dependent apoptosis [29]. These reports suggest that antioxidant peptides may be critical to the antioxidant capacity of MFGM.

### 3.4. Pearson Correlation Analysis of Different Indicators

The relationship between blood biomarkers and antifatigue measurements was analyzed via Pearson correlation analysis. Changes in these parameters, such as EST, BG, BLA, and LDH, reflect the biochemical status of body fatigue. As shown in Figure 3, the levels of these fatigue-associated indices are highly correlated with antioxidant enzymes (SOD, CAT, and GSH-Px) and are inversely related to the MDA content.

These observations are supported by many other studies. For example, Chen et al. reported that feeding BALB/c mice quercetin, an antioxidant belonging to polyphenol, significantly increased swimming time and reduced BLA and LDH [30]. Peptides from the digestion of MFGM protein may also have antifatigue properties. The MFGM in this study contained 67% of protein, and their digestion might produce antifatigue peptides. For example, the peptides from the protein hydrolysate of large-head hairtail (*Trichiurus lepturus*), a member of the cuttlefish family, improved the exhaustive swimming time of mice gavaged with these peptides for six weeks. The Pearson correlation analysis of fatigue-related indices of the hairtail peptide feeding, including metabolites and energy-related indices, were significantly correlated with antioxidant levels [31]. Another example of the importance of ROS reduction in decreasing fatigue is the study by Zhu and coworkers, who found that macamides, composed of long-chain fatty acid and benzylamine from Maca (*Lepidium meyenii* Walp.)*,* decrease the ROS levels in the serum and muscles of mice after a swimming test and up-regulate the expression of heme oxygenase-1, which belongs to the Nrf2 pathway, in the liver. Their Pearson correlation analysis showed a high correlation between fatigue-related indices, antioxidant enzymes, and ROS levels [32]. These studies suggest that antioxidant capacity plays a crucial role in the antifatigue effects of MFGM. However, further research is necessary to reveal the detailed mechanisms by which MFGM, a substance containing antioxidants, proteins, and lipids, relieves fatigue.

### 3.5. Effects of MFGM on Protein Expressions in Muscle and Liver

The nuclear factor erythroid 2-related factor 2 (Nrf2) signaling pathway plays a crucial role in resisting oxidative stimulation and maintaining the homeostasis of redox status in tissues and organs. To further clarify the antioxidation and fatigue-resisting mechanisms of MFGM, we analyzed the protein levels of antioxidant proteins Nrf2 and Keap1 (Kelch-like ECH-associated protein 1). The relative protein expressions of total and nuclear Nrf2 in the MFGM group are significantly higher than those in the Control group (*p* < 0.05) (Figure 4A,B).

Under normal homeostatic circumstances, Nrf2 is anchored by both Keap1 and the actin cytoskeleton, and it is then degraded in the cytoplasm. When oxidative stress happens, Nrf2 dissociates from Keap1 and enters the nucleus to maintain redox homeostasis through the regulation of downstream proteins. Oh et al. generated an *Nrf2* knockout mouse model and reported that wild-type mice had longer running distances than *Nrf2*-null mice in an exhaustive treadmill test. During the exhaustive exercise, the *Nrf2*-null mice suffered more severe oxidative stress damage and decreased motor function than the wild-type mice [33]. The MFGM used in this study contained 67% of protein, which may have functional properties. Wang et al. reported that sea cucumber peptides (SCP) significantly improved exercise performance and fatigue-related physiological indicators, increased the activity of antioxidant enzymes, and inhibited the free radical metabolite, MDA, in the serum of mice. SCP regulated oxidative stress and exerted an antifatigue effect by regulating the Nrf2 pathway [34]. These results are similar to the results of the current study, suggesting that MFGM regulation of the Nrf2 pathway, especially the expression of Nrf2, improves antifatigue capacity by maintaining redox homeostasis. It is worth noting that there is no significant difference in the expression of Keap1 and cytoplasmic Nrf2 between the Control and MFGM groups (Figure 4A,B). ETGE, a stretch of four amino acids within the N-terminal region of Nrf2, is a vital motif for the Nrf2–Keap1 interaction. The loss of the ETGE motif’s function was identified as abolishing the repressive effect of Keap1 on Nrf2 [35]. ETGE motif might be influenced after the supplementation of MFGM, destroying the interaction of Nrf2 and Keap1. Afterward, the degradation of Keap1 and cytoplasmic Nrf2 becomes disordered.

The cleaved Caspase-3 expression is significantly decreased in the gastrocnemius muscle of the mice after MFGM supplementation (Figure 4C,D). As a critical factor for inducing apoptosis, the level of cleaved caspase-3 has a positive correlation with the degree of apoptosis [36]. The apoptosis pathways are sensitive to and are regulated by the intracellular redox environment. Mitochondria-generated ROS have a positive effect on the release of pro-apoptotic proteins [37]. The skeletal muscle’s nuclear and mitochondrial integrity is damaged by ROS, causing fatigue. Bcl2 and Bax are other crucial apoptosis regulators, with Bcl2 being anti-apoptosis and Bcl-2-associated X protein (Bax) being pro-apoptosis. A previous study reported that Nrf2 inhibitors decreased the expression levels of Bcl2 and the Bcl2/Bax ratio, which then increased the expression level of downstream Caspase-3 [38]. Ma et al. showed that the polyphenol, phlorizin (PHZ), increased the exhaustive exercise-induced fatigue time in mice. They found that PHZ facilitated Nrf2 translocation from the cytoplasm to the nucleus, increased the ratio of Bcl2/Bax, and decreased cleaved Caspase-3 [39]. These results demonstrate that the Nrf2 pathway can be mediated by foods and suggest that MFGM might also relieve fatigue via targeting Nrf2 to inhibit apoptosis and protect skeletal muscle.

Besides the pathways mentioned above, the JNK (c-Jun amino-terminal kinase) pathway also deserves attention. As a potent activator for JNK, ROS inactivates endogenous JNK inhibitors oxidatively [40]. JNK induces Nrf2-Keap1 dissociation, Nrf2 upregulation, and Nrf2 movement to the nucleus to relieve oxidative stress [41]. In addition, JNK could phosphorylate the proapoptotic Bcl-2 family protein BAD to suppress apoptosis [42], which greatly corresponds with the lower expression level of cleaved Caspase-3 in the MFGM group. Previous studies reported that protocatechuic acid induced antioxidant enzyme expression and prevented oxLDL-induced apoptosis through JNK-mediated Nrf2 activation in murine macrophages [43]. Yuan and coworkers declaimed that paeoniflorin could significantly inhibit cell apoptosis and resist H_2_O_2_-induced oxidative stress in melanocytes through the JNK/Nrf2 pathway [44]. Nevertheless, to verify the above conjecture more rigorously, the *Nfe2l2*, *Keap1*, and *Mapk8* knockout mouse should be used to clarify the mechanism of MFGM’s antifatigue effects in future studies.

### 3.6. Effects of MFGM on the Relative Abundance and Composition of Gut Microbiota

The Shannon indices, related to the alpha-diversity of the gut microbiome, between the Control and MFGM groups (Figure 5A) are similar, indicating that the supplementation of MFGM did not alter the diversity of the gut microbiota. However, the principal component analysis (PCA, Figure 5B) shows that the taxonomic composition differs between the two groups. As revealed in Figure 5C–E, MFGM results in advantageous changes in the gut microbiota from the phylum to the genus level. Although the changes in the microbiota within the group are not completely consistent due to individual differences, we mainly studied the bacteria that have varied as a whole between the groups. At the phylum level, the main compositions of the microbial community are Bacteroidota, Firmicutes, Proteobacteria, Verrucomicrobiota, and Actinobacteriota, accounting for more than 98% of the total. Among them, the relative abundance of the phylum Verrucomicrobiota significantly increases to 2.25% (*p* < 0.05) after the MFGM supplementation. In a previous study, an increase in Verrucomicrobiota in C57bl/6 mice fed a six amino acid peptide from Jinhua ham was associated with decreased MDA, ROS, and antioxidant enzymes [45]. The peptide benefits were attributed to increased tight junction protein and reduced lipopolysaccharide (LPS)-induced inflammation. The relative abundance of Firmicutes is increased in the mice of the MFGM group. Matsumoto et al. reported that increased phylum Firmicutes (SM7/11 and T2-87) produced greater colonic butyrate in rats participating in a voluntary running exercise [46]. To further characterize the gut microbiota changes, besides observing the gut microbiota in each mouse in Figure 5C–E, the LEfSe analysis was used to distinguish the taxa with significant differences in abundance (LDA score > 3.0, *p* < 0.05, Figure 6F,G).

At the family level, there is an increased relative abundance of Bacteroidaceae and decreased relative abundance of Helicobacteraceae after the MFGM supplementation. Bacteroidaceae produce short-chain fatty acids (SCFAs) [47]. Carbohydrates are digested and subsequently fermented into SCFAs in the colon, where they are carried through the bloodstream to various organs (e.g., muscle and adipose tissues) and used as substrates for gluconeogenesis and lipogenesis [48]. In a BALB/c allergen mouse model, reduction of the Helicobacteraceae family (Proteobacteria phylum) reduced inflammatory biomarkers by disrupting the recycling of tight junction proteins that form the physical and immunologic barrier of the intestinal epithelium [49]. Disruption of the tight junction of the mucosal epithelial cells may damage the water transport and mucosal hydration function. The decreased hydration status of the intestine may reduce endurance performance [6].

The relative abundance of *Bacteroides*, *Butyricimonas*, and *Anaerostipes* increases after the MFGM supplementation at the genus level. The MFGM reduces the relative abundance of *Lactobacillus* compared to the Control group. *Bacteroides* play a significant role in producing SCFAs, mainly propionate and acetate, to satisfy the energy requirements in endurance exercises [50]. Meanwhile, members of *Bacteroides* are oxygen tolerant and can survive oxidative stress when the host faces high-intensity exercise [51]. As a genus predominant in the gut of endurance athletes [52], *Lactobacillus* can promote the utilization of protein and maintain the homeostasis of energy metabolism in muscle. However, the relative abundance of *Lactobacillus* decreases in the mice of the MFGM group. This contradiction might be due to the production of tryptophan by Lactobacillus and the subsequent synthesis of 5-HT in the gut. 5-HT transported to the brain affects the pituitary gland and mental mood, thereby resulting in central fatigue [7]. *Butyricimonas* and *Anaerostipes* are the main genera that produce butyrate, which regulates the neutrophil function and migration, and increases the expression of tight junction proteins in colon epithelia. Butyrate can be utilized as an important source of energy for other microbes and the host’s organs [6]. The current study’s results show that MFGM reshapes the gut microbiome of mice associated with antifatigue capacity. However, further research is needed to clarify the detailed mechanism.

### 3.7. Effects of MFGM on the Functional Changes in Microbial Communities

To gain insight into the shifts of molecular functions of bacterial microbiota in response to the changes in the gut microbial communities, the KEGG analysis was performed via a phylogenetic investigation of the community through the reconstruction of unobserved states (PICRUSt). As presented in Figure 6, Genetic Information Processing, Organismal Systems, Human Disease, Cellular Processes, Environmental Information Processing, and Metabolism exhibit significant alternations at level 1 of the KEGG after the MFGM supplementation (*p* < 0.05). Especially at level 3 in Metabolism (Figure 6F), it is observed that some pathways associated with increased antifatigue capacity are enriched in the MFGM group.

In lipid metabolism, sphingolipid metabolism and glycerophospholipid metabolism are enriched after the MFGM supplementation. In previous research, sphingolipid metabolism was significantly altered after exercise in rats with chronic fatigue syndrome [53]. Sphingolipid was also demonstrated to play a critical role in muscle contraction and protection from fatigue [54]. As bioactive molecules are involved in the recognition and signal transduction of proteins, glycerophospholipid is one of the phospholipids in the cell membrane. He and coworkers declared that combining Astragali Radix and Codonopsis Radix with Jujubae Fructus could protect cell membrane structure from oxidative damage and ameliorate fatigue by regulating glycerophospholipid metabolism in mice [55]. In amino acid metabolism, the metabolism of BCAAs increases in the MFGM group. As mentioned above, decreasing the free Trp/BCAA ratio can reduce the synthesis and release of 5-HT in neurons, thereby delaying central fatigue [25]. The proportion of BCAAs reaching 19.73% in the MFGM group may be the reason for the increase in BCAA metabolism. D-Arginine and D-ornithine metabolisms are higher in the MFGM group. In a previous study, the supplementation of arginine and ornithine increased the EST of rats, and the potential mechanism was associated with ammonia buffering [56]. In terms of vitamin metabolism, MFGM significantly promotes the biosynthesis and metabolism of various B vitamins, for example, folate, biotin, and vitamin B6. B vitamins have been suggested to directly stimulate ROS and modulate immune cytokines to reduce oxidant stress [57]. Vitamin B complex has been claimed to relieve oxidative tissue injury related to stress-induced neurobehavioral changes in rats [58]. Moreover, the apoptosis pathway of Cellular Processes is significantly increased in the Control group, which is similar to the result obtained from the immunohistochemical analysis of gastrocnemius muscle in Figure 4. However, a more in-depth study is needed to confirm the potential relationship between MFGM and the abovementioned pathway.

## 4. Conclusions

The present study demonstrated that chronic MFGM supplementation for 18 weeks significantly increased the antifatigue capacity of mice. According to the high correlation between antioxidant enzymes and fatigue-related indices, as well as the increased level of Nrf2 in the liver after the MFGM supplementation, this effect might be related to the regulation of the Nrf2 pathway, which is associated with resistance against oxidative stress, thereby relieving fatigue. MFGM improves gut bacterial composition at the phylum, family, and genus levels, while altering microbiota function associated with antifatigue capacity. This study expands the application fields of MFGM and provides a theoretical basis for the development and application of new antifatigue products.

## Figures and Tables

**Figure 1 antioxidants-12-00712-f001:**
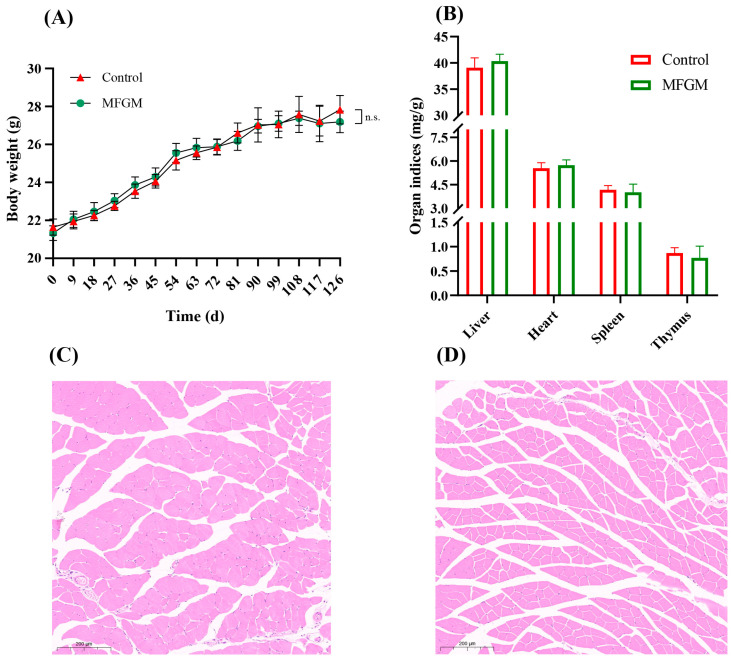
Effects of milk fat globule membrane on body weight, organ indices, and muscle tissue: (**A**) body weight of mice in different groups over 18 weeks; (**B**) organ indices of mice in different groups at 18 weeks; (**C**) the structure of gastrocnemius in the Control group; and (**D**) the structure of gastrocnemius in the MFGM group. The data are expressed as mean ± SD, *n* = 8, n.s.: no significance. Control: fed with saline; MFGM: fed with 400 mg/kg BW milk fat globule membrane.

**Figure 2 antioxidants-12-00712-f002:**
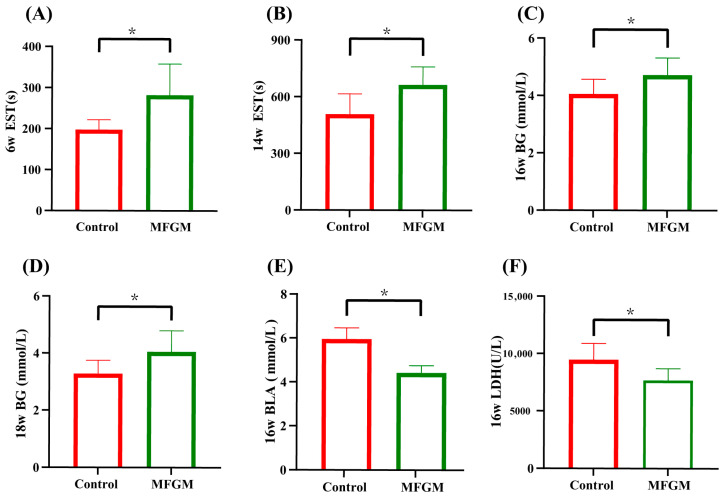
Effects of milk fat globule membrane on the swimming test and biochemical indices in blood serum: (**A**) exhaustive swimming time (EST) of the mice in different groups at 6 weeks; (**B**) exhaustive swimming time (EST) of the mice in different groups at 14 weeks; (**C**) content of blood glucose (BG) from the mice in different groups after the free swimming test at 16 weeks; (**D**) content of blood glucose (BG) from the mice in different groups at 18 weeks; (**E**) content of blood lactic acid (BLA) from the mice in different groups after the free swimming test at 16 weeks; and (**F**) content of lactate dehydrogenase (LDH) from the mice in different groups after the free swimming test at 16 weeks. The data are expressed as mean ± SD, *n* = 8, * *p* < 0.05. Control: fed with saline; MFGM: fed with 400 mg/kg BW milk fat globule membrane.

**Figure 3 antioxidants-12-00712-f003:**
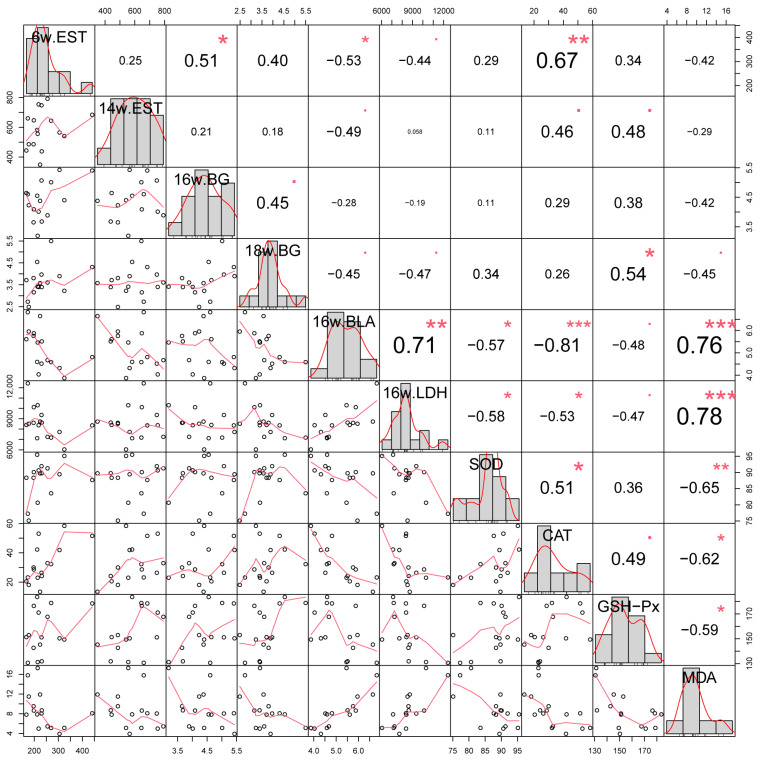
Pearson correlation analysis of different indicators of mice. 6w. EST: exhaustive swimming time at 6 weeks; 14w. EST: exhaustive swimming time at 14 weeks; 16w. BG: blood glucose after free swimming test at 16 weeks; 18w. BG: blood glucose at 18 weeks; 16w. LDH: lactate dehydrogenase after free swimming test at 16 weeks; SOD: superoxide dismutase at 18 weeks; CAT: catalase at 18 weeks; GSH-Px: glutathione peroxidase at 18 weeks; and MDA: malondialdehyde at 18 weeks. All biochemical indices are measured from blood serum. The red line at the bottom half of the plot is the fitted curve. The red line in the diagonal of the plot is the probability density function. Significant correlations are indicated by * *p* < 0.05, ** *p* < 0.01, and *** *p* < 0.001.

**Figure 4 antioxidants-12-00712-f004:**
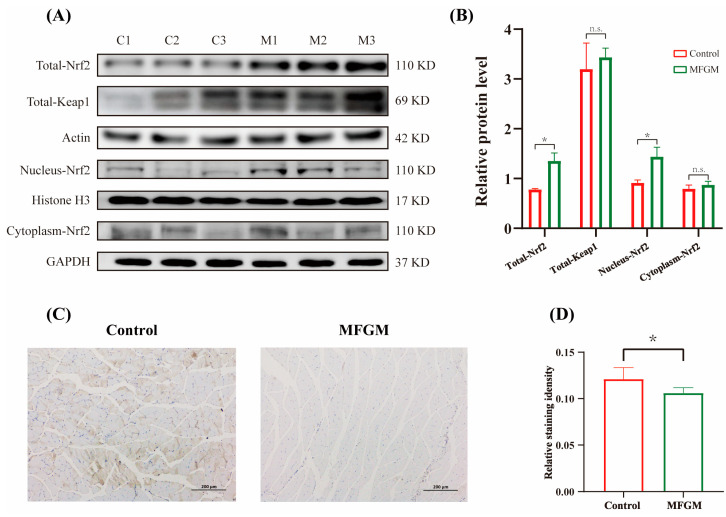
Effects of milk fat globule membrane on protein expressions in the muscle and liver of the mice at 18 weeks: (**A**) Western blot of nuclear and cytoplasmic nuclear factor erythroid 2-related factor 2 (Nrf2) and total Kelch-like ECH-associated protein 1 (Keap1); (**B**) quantification analysis of total Nrf2, total Keap1, nucleus-Nrf2, and cytoplasm-Nrf2; (**C**) immunohistochemical analysis of gastrocnemius muscle in the Control and MFGM groups; and (**D**) average optical density of Caspase-3 in the two groups. C1, C2, and C3 belong to the Control group, whereas M1, M2, and M3 belong to the MFGM group. The data are expressed as mean ± SD, *n* = 3, * *p* < 0.05, n.s.: no significance. Control: fed with saline; MFGM: fed with 400 mg/kg BW milk fat globule membrane.

**Figure 5 antioxidants-12-00712-f005:**
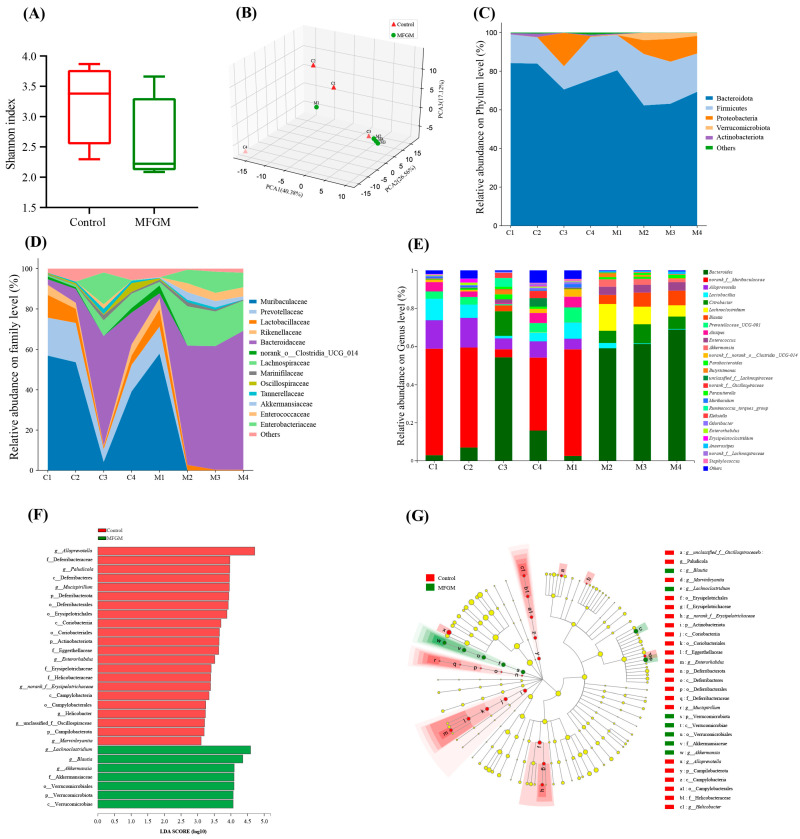
Effects of milk fat globule membrane on intestinal flora at 18 weeks: (**A**) Alpha-diversity was assessed by calculating the Shannon index; (**B**) beta-diversity was assessed using principal component analysis; (**C**) effects of milk fat globule membrane on the composition of gut microbiota in the mice (Phylum level); (**D**) effects of milk fat globule membrane on the composition of gut microbiota in the mice (Family level); (**E**) effects of milk fat globule membrane on the composition of gut microbiota in the mice (Genus level); (**F**) results of linear discriminative analysis (LDA); and (**G**) effect size (LefSe) analysis between the two groups. Cardiogram showing differentially abundant taxonomic clades with an LDA score of 3.0 among groups with a *p*-value of 0.05; C1, C2, C3, and C4 belong to the Control group, whereas M1, M2, M3, and M4 belong to the MFGM group. Control: fed with saline; MFGM: fed with 400 mg/kg BW milk fat globule membrane.

**Figure 6 antioxidants-12-00712-f006:**
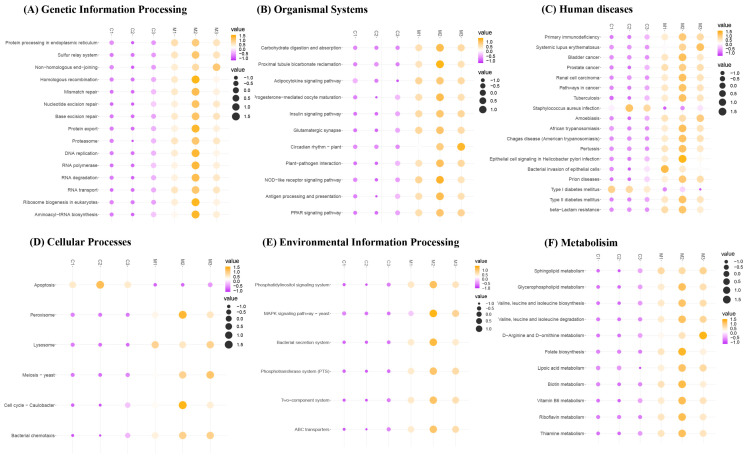
The predicted functional differences based on the KEGG (Kyoto Encyclopedia of Genes and Genomes) of the metabolic pathways of the gut microbiota at 18 weeks: (**A**) Genetic Information Processing; (**B**) Organismal Systems; (**C**) Human Diseases; (**D**) Cellular Processes; (**E**) Environmental Information Processing; and (**F**) Metabolism. C1, C2, and C3 belong to the Control group, whereas M1, M2, and M3 belong to the MFGM group. Control: fed with saline; MFGM: fed with 400 mg/kg BW milk fat globule membrane.

**Table 1 antioxidants-12-00712-t001:** The proportion of phospholipids in milk fat globule membrane.

Content	Proportion (%)
Phosphatidylcholine (PC)	1.97
Sphingomyelin (SM)	1.86
Phosphatidylethanolamine (PE)	2.61
Phosphatidylinositol (PI)	0.14
Phosphatidylserine (PS)	0.88
Total	7.46

**Table 2 antioxidants-12-00712-t002:** Effects of milk fat globule membrane on antioxidant indices in the blood serum of the mice at 18 weeks.

	SOD	CAT	GSH-Px	MDA
	(U/mL)	(U/mL)	(μmol/L)	(nmol/mL)
Control	84.92 ± 6.20	22.61 ± 5.39	145.33 ± 15.40	11.40 ± 3.50
MFGM	90.62 ± 3.89 *	42.32 ± 11.36 *	166.28 ± 14.10 *	6.07 ± 1.66 *

SOD, superoxide dismutase; CAT, catalase; GSH-Px, glutathione peroxidase; MDA, malondialdehyde. The data are expressed as mean ± SD, *n* = 8, * *p* < 0.05. Control: fed with saline; MFGM: fed with 400 mg/kg BW milk fat globule membrane.

## Data Availability

Data are contained within the article and Appendix A.

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
