# Peer review of "Milk Fat Globule Membrane Relieves Fatigue via Regulation of Oxidative Stress and Gut Microbiota in BALB/c Mice"

_antioxidants, 2023, doi:10.3390/antiox12030712_

Round 1

Reviewer 1 Report

Regarding MS entitled ‘’ Milk Fat Globule Membrane Relieved Fatigue via Regulation 2 of Oxidative Stress and Gut Microbiota in BALB/c Mice’’ This manuscript is interesting, however, I have some comments.

Abstract.

Please add more information about the experimental design.

Add p-value for significant findings

L25. The conclusion should be strong and with a recommendation to the readers.

Introduction

Well-written introduction, please L66. Add hypothesis

Materials and Methods

Informative and well-structured

L103. Add ref, for the selected dose.

Statistical analysis, the authors used one-way ANOVA, however, they have only two treatments.  One-way ANOVA is used for more than two treatments.

The results and discussion section is long please shorten as possible.

Figure 5, please magnify it to be clear for the readers.

Reviewer 2 Report

Dear Editor,

Thank you for giving me the possibility to read the paper entitled Milk Fat Globule Membrane Relieved Fatigue via Regulation of Oxidative Stress and Gut Microbiota in BALB/c Mice (antioxidants-2145420).

The paper is well written and falls within the topics of the journal. I think it could be accepted after minor revision, suggesting just minor changes to the material and methods section in order to be clearer for the reader.

L69 to 72 – The chemical composition dose not reach the 100%, please check the values and list other calculated components if present. Moreover, I tink that the techniques used by the AOAC methods should be mentioned.

L186 to191 – It is the only statistical analysis performed? Is the same for all the investigated parameters? Maybe that some of them should be analysed in different ways. For example body weight. Please check and improve this section.

Reviewer 3 Report

Manuscript ID: antioxidants-2145420

Milk Fat Globule Membrane Relieved Fatigue via Regulation of Oxidative Stress and Gut Microbiota in BALB/c Mice

This is an interesting paper investigating the effect of MFGM on fatigue by the regulation of oxidative stress and gut microbiota.

One aspect of the paper that I found confusing was where the authors results ended and the discussion started. It is often only the reference at the end of the sentence that indicates it is not the author’s work. I would suggest that if the results and discussion are in the same section that the discussion start in a new paragraph to alert the reader.

For instance, lines 202, 230, 294, 343, 377, 389, 401, 414, 445.

This also highlights the fact that the results are not commented on in great detail, often only referring to a figure and asking the reader to absorb what it means. There should be more explanation of the results.

The other concern is the diets. The basal diets seemed to have been ignored and only the supplementation diets discussed. The control was saline and the treated was Hilmar 7500 which is a whey product with 7.46% phospholipids. The effect of branch chain amino acids is discussed in line 247. While the BCAA in the MFGM are discussed, there is no information on how much BCAA there is in the base diet and hence what proportion the MFGM BCAA is of the total diet and whether this would be significantly different from the control. (Supplementary data was not included in the review manuscript.) Rational for differences between the control and treatment groups should be based on the whole diet and not just the supplemented diet differences.

Minor comments

L203 could also state the MFG-8 alternatively known as lactadherin.

Figure 3. Explain what the red lines are
Figure 3. Are these all the mice (control and treatment) ?

Line 237 What was the source from which the antioxidant proteins were measured?

Line 330 Should be figure 4 not figure 3

Figure 5 D and E, Authors should discuss variability within the control and MFGM groups

Round 2

Reviewer 3 Report

The authors have addressed all points except the BCAA in the total diet.

Their response was "Because the maintenance feed was eaten by all mice in Control and MFGM groups every day, we could think that the antifatigue effects of BCAAs in basic diets canceled each other out".  

This is true, but the reader would wish to know how much larger the BCAA was in the supplemented diet to know the extent of the impact.

As a compromise, in line 102 I would like the authors to add the product number of the maintenance feed from Beijing Keao Xieli Feed Co., Ltd (Beijing, China) and also to state the type of protein used (e.g. casein).

This would allow the readers to find the exact composition if required.

Also, this would give the reader some knowledge of the background feed, given that the supplementary feed was whey proteins.

If this was done, I would accept without the need for further review.
